# Photonics of Some Monomethine Cyanine Dyes in Solutions and in Complexes with Biomolecules

**DOI:** 10.3390/ijms241813954

**Published:** 2023-09-11

**Authors:** Pavel G. Pronkin, Alexander S. Tatikolov

**Affiliations:** N.M. Emanuel Institute of Biochemical Physics, Russian Academy of Sciences, 4 Kosygin Str., 119334 Moscow, Russia; pronkinp@gmail.com

**Keywords:** monomethine cyanine dyes, spectral fluorescent study, DNA, human serum albumin, fluorescent probes, fluorescence growth, complexation, triplet state

## Abstract

In search of new probes for biomolecules, the spectral fluorescent study of four monomethine cyanine dyes (MCD), both unsymmetrical and symmetrical, has been carried out in different organic solvents, in aqueous buffer solutions, and in the presence of DNA and HSA. The complexation of MCD with biomacromolecules leads to a steep growth of the fluorescence intensity. Complexes of MCD with dsDNA and HSA of various types were modeled in silico by molecular docking. Experiments on thermal dissociation of dsDNA in the presence of MCD showed the formation of intercalative complexes of MCD with DNA. Quenching of intrinsic fluorescence of HSA by MCD occurred with rate constants much higher than the diffusion limit, that is, in dye–HSA complexes. Effective constants of MCD complexation with the biomacromolecules were estimated. MCD **1** has the best characteristics as a possible fluorescent probe for dsDNA and can serve as a sensitive and selective probe for dsDNA in the presence of HSA. Photochemical properties of MCD complexed with DNA have been also studied. An increase in the quantum yield of the triplet states of MCD in complexes with DNA has been found, which may be important for using these dyes as potential candidates in photodynamic therapy.

## 1. Introduction

Various biochemical methods are used for the detection and quantification of biomacromolecules (e.g., for DNA—PCR analysis, DNA sequencing, gel electrophoresis, etc.); however, those methods often need sophisticated and expensive equipment and suffer from complicated and time-consuming procedures. An alternative for these purposes is the use of dye probes and sensors, which gives advantages both in speed and simplicity of the method [1,2,3]. Furthermore, modern dye probes can provide sufficiently high sensitivity and selectivity toward target biomolecules [4]. Polymethine (cyanine) dyes can be used as such probes due to their unique spectral fluorescent properties (first of all, the dependence of their properties on the molecular environment) [5,6]. Monomethine cyanine dyes (MCD), containing a polymethine chain with only one methine unit, effectively interact with DNA and other nucleic acids, which led to their use for the detection and staining of nucleic acids in various biological objects. In particular, MCD thiazole orange [7] and oxazole yellow [8] are widely used for the detection and staining of nucleic acids; SYBR Green I and II are used for the same purposes (see, for example, [9]), and PicoGreen allows detection of nanomolar concentrations of DNA in solution [10,11]. MCDs are also used in real-time PCR analysis of nucleic acids [12,13,14,15]. A wide range of biological and biomedical systems currently being studied with the use of spectral fluorescent probes stimulates a permanent emergence of research aimed at developing new efficient and selective dye probes for the detection and staining of biomolecules in such systems (see, e.g., [16,17,18]). There are a large number of works by different research groups devoted to the synthesis and spectral fluorescent study of various MCDs (symmetrical and unsymmetrical) in order to develop probes for nucleic acids [16,17,18,19,20,21,22,23,24,25,26,27,28,29,30,31,32,33,34]. There are also some works directed to the study of primary photophysical and photochemical processes in MCD molecules, both free and noncovalently bound to nucleic acids [35,36,37,38,39,40] (see also the review [41]). Such studies can give a new impetus to the application of MCD in biological objects. In the recent work [26], in search of new fluorescent probes for nucleic acids, a number of unsymmetrical MCDs were synthesized, which interacted with RNA and DNA with fluorescence growth. However, their photophysical properties were studied in [26] very briefly. In the present work, we have undertaken a comprehensive study of the spectral fluorescent properties of two of the previously synthesized [26] unsymmetrical MCDs, **1** and **2**, in various solvents, their noncovalent interaction with dsDNA, ssDNA, and human serum albumin (HSA), and photonics (spectral fluorescent and photochemical properties) of dye–biomolecule complexes with a view to the possible use of these dyes as probes in biomolecular systems. Symmetrical MCD **3** and **4**, having terminal heterocycles similar to those of **1** and **2**, were also studied for comparison. The structures of the studied dyes are shown in Figure 1.

## 2. Results and Discussion

### 2.1. Solvatofluorochromism of MCD **1**–**4** in Organic Solvents and Aqueous Medium

Solvatofluorochromic properties of MCD **1**–**4** were studied in organic solvents of different polarities (see Table 1). The absorption spectra of MCD **1**–**4** represent single bands of π–π* transitions (S_0_–S_1_); for MCD **4**, an additional short-wavelength vibronic band/shoulder (with Δλ_max_~25–30 nm) is observed. The absorption spectra of MCD **1** in some solvents are shown in Figure 2A.

In organic solvents, the fluorescence spectra of MCD **1**–**3** represent low-intensity single bands (for MCD **1** see Figure 2B); the dyes are characterized by low fluorescence quantum yields (φ_fl_; see Table 2; fluorescence of MCD **4** is extremely weak). For MCD **1** and **3**, the Stokes shift of the band maxima is Δλ_St_ = 30–58 nm, depending on the solvent. For MCD **2** it is smaller (Δλ_St_ = 20–35 nm). The fluorescence excitation spectra of MCD, along with a main band, exhibit a characteristic rise in the short-wavelength region (< 350 nm), which is possibly associated with the contribution of electronic transitions in terminal heteronuclei (S_0_–S_n_) (for MCD **1** see Figure 2C). The maxima of the fluorescence excitation spectra are similar to those of the absorption spectra (the difference in the maxima Δλ = 2–6 nm).

The rate constants of the radiative (*k*_r_) and nonradiative (*k*_nr_) deactivation of the excited state and the radiative lifetimes (τ_r_) were determined from the spectral data; in polar solvents τ_r_ = 2.1–5.7 ns (Table 2).

In weakly polar (ethyl acetate) and nonpolar (1,4-dioxane) media, the observed extinction coefficients (ε) of the dyes decrease. In particular, for unsymmetrical MCD **1** and **2** in nonpolar 1,4-dioxane, compared to solutions in ethanol, the maximum absorbance (*A*_max_) decreased almost 3 and 1.7 times, respectively. For MCD **3** and **4** under the same conditions, the decrease in *A*_max_ was 27% and 43%, respectively. Using the method of spectral moments, it has been shown that the width of the absorption spectra of MCD **1** (σ_abs_) in nonpolar 1,4-dioxane and in low-polarity ethyl acetate is approximately 10–40% larger than in polar DMSO and ethanol. In weakly polar and nonpolar media, some increase in φ_fl_ was found for MCD **1**–**3** (for MCD **1**, an increase in φ_fl_ was ~3.6 times; Table 2). This is probably due to the formation of dye–counterion ion pairs in nonpolar solutions [43]. Under these conditions, intramolecular processes of nonradiative deactivation of the electronic excitation energy in MCD molecules are partially hindered (a decrease in *k*_nr_ is observed; see Table 2), which can lead to an increase in the fluorescence intensity *I*_fl_ (φ_fl_). Note that in more viscous solvents, *I*_fl_ of MCD is somewhat higher [27]. For MCD **4** in 1,4-dioxane and ethyl acetate, the appearance of additional short-wavelength bands in the fluorescence and fluorescence excitation spectra (see Table 1) can be explained by the formation of fluorescent associates (aggregates).

The influence of various solvent parameters such as the dielectric constant (relative permittivity) ε_r_, refractive index *n*, polarizability *P** = (*n*^2^ – 1)/(*n*^2^ + 2), *E*_T_(30) of Reichardt [44], the solvent polarity parameter *Z* of Kosower, the general basicity (nucleophilicity) parameter of solvents *B* [44,45], on the spectra of MCD **1**–**4** was studied. Table 3 for MCD **1** and **2** presents the values of the first moment of the normalized distribution of photons (ν_1_), the width (σ) of the absorption spectra (obtained by the method of moments [46]), and the fluorescence intensity (*I*_fl_). The data for MCD **3** and **4** are presented in Appendix A (Appendix A).

In polar solvents, very weak correlations were observed between λ_abs_ or ν_1_ and the refractive index *n* or the polarizability *P** of the solvent. For MCD **1** and **2**, the correlation coefficients *R*^2^ = 0.51–0.7 (for linear dependences of λ_abs_ on *n* or (*n*^2^ − 1)/(*n*^2^ + 2)) were obtained. Very small solvatochromic shifts were observed earlier for other monomethine cyanines [27]. All these data suggest that the electronic distribution in the dye molecule is practically unaffected by the solvent polarity/polarizability [27].

In aqueous solutions (PBS buffer, at *c*_MCD_ ≤ 0.6–1 × 10^−5^ mol L^−1^), the absorption spectra do not exhibit a significant influence of aggregation of MCD **1**–**4**: neither new bands appear nor a noticeable increase in σ_abs_ is observed in comparison with their spectra obtained in polar organic solvents (ethanol, DMSO). However, in an aqueous medium, the observed molar extinction coefficients of the dyes decrease somewhat. In particular, for MCD **3** and **4**, we determined ε~61,800 and 44,500 L mol^−1^ cm^−1^, which are less than the ε values in ethanol (69,200 and 49,000 L mol^−1^ cm^−1^, respectively [42]). In the fluorescence excitation spectra of MCD **1**, **3**, **4** in PBS, additional bands are observed, hypsochromically shifted relative to the main bands (Δλ_ex_ = 23–70 nm, Table 1), which can be attributed to the fluorescence of H-aggregates. In the case of MCD **2**, only one band is observed in both absorption and fluorescence excitation spectra, which indicates the absence of aggregates for this dye. The fluorescence intensity of H-aggregates of the dyes is comparable with the fluorescence intensity of their monomers in solution: in particular, for MCD **1** in PBS, the total quantum efficiency of fluorescence is φ_fl_~0.05%, whereas for MCD **2** φ_fl_~0.03%).

The ability of MCD to self-aggregate in solutions is determined by the lipophilic (hydrophobic) properties of the dye molecules. For a qualitative assessment of the lipophilicity of the studied dyes, the calculated values of the n-octanol–water partition ratio (*K*_ow_) were used [47,48]. For MCD **1**, **3**, **4**, *K*_ow_ values range from 15 to 450, indicating essential lipophilic properties. The presence of a pyridinium fragment lowered the *K*_ow_ value for MCD **2** by many orders of magnitude: milogP = −4.63, whereas for MCD **1**, **3**, and **4** milogP = 2.67, 1.19, and 1.82, respectively (for example, for 1-pentylpyridinium milogP = −2.53). This indicates the high hydrophilicity of MCD **2**, which explains the absence of a tendency to aggregate in aqueous solutions.

### 2.2. Influence of Biopolymers on Spectral Fluorescent Properties of Cyanine Dyes

Complexation of MCD **1**–**4** with DNA (double-stranded and single-stranded) and with human serum albumin (HSA) was studied by spectral fluorescent methods in a wide range of biopolymer concentrations (*c*_dsDNA_ = 0–1.5 × 10^−4^ mol L^−1^; *c*_ssDNA_ = 0–1.4 × 10^−4^ mol L^−1^; *c*_HSA_ = 0–1.4 × 10^−4^ mol L^−1^ in PBS buffer). The absorption, fluorescence, and fluorescence excitation spectra of MCD **1** and **3** in the presence of dsDNA are shown in Figure 3 and Figure 4, respectively.

At relatively low biopolymer concentrations *c*_rel_ = *c*_dsDNA_/*c*_MCD_ ≤ 0.5–0.6, a slight increase in the band intensity is observed in the MCD absorption spectra (~10%; for MCD **1** see Figure 3A, curves *1* and *2*). With a further increase in *c*_dsDNA_, the spectra of MCD **1**–**4** show a decrease in the apparent extinction coefficients (hypochromia) and an increase in the bathochromic shifts of the absorption bands (see Figure 3A and Figure 4A; the shifts of the Δλ_absDNA_ bands are shown in Table 4). For MCD **3** (see Figure 4A) and **4** at *c*_dsDNA_ = 1 × 10^−4^ mol L^−1^, the hypochromia is 44% and 36%, respectively, compared to the initial spectra in PBS. For MCD **1** and **2** at high concentrations of dsDNA, the hypochromia was 8% and 17%, and Δλ_abs_~13 and 7 nm, respectively. No new bands, which could be assigned to ordered H- or J-aggregates of the dyes, appear in the absorption spectra.

The method of spectral moments made it possible to expand the number of parameters for assessing changes in the absorption spectra of MCD **1**–**4** in the presence of biopolymers. The processing of the spectra has shown that at 0.6 < *c*_rel_ ≤ 4–5, the interaction with dsDNA leads to the shifts of the “centers of gravity” of the MCD **1**–**3** spectral bands: for MCD **1**, depending on the concentration of dsDNA, a long-wavelength shift is observed (on the wavelength scale, Δ*M*^−1^~4–5 nm). A further increase in *c*_rel_ leads to an increase in this effect: for MCD **1** and **2**, Δ*M*^−1^~8–9 nm; for MCD **3** and **4**, the changes are weaker (see Table 4). An increase in *c*_rel_ has different effects on the width of the absorption spectra of MCD **1**–**4** (σ_abs_): for MCD **1** and **2**, the interaction with dsDNA reduces σ_abs_ by 6–7% (at *c*_dsDNA_~1 × 10^−4^ mol L^−1^), while for symmetrical MCD **3** and **4**, an increase in σ_abs_ is observed.

Such changes in the absorption spectra are due to changes in the spectral characteristics of the free and bound dyes: MCD bound to DNA have apparently lower extinction coefficients and somewhat longer-wavelength absorption spectra than the unbound dyes.

Complexation with DNA has a striking effect on MCD fluorescence—a steep growth of the fluorescence intensity. In the presence of dsDNA, the fluorescence excitation spectra of MCD **1**–**4** represent single bands (Figure 3C and Figure 4C; Appendix A in Appendix A) corresponding to bound monomeric forms (with no contribution of aggregates). The band maxima in the fluorescence spectra of the dyes undergo multidirectional shifts (see Figure 3B and Figure 4B; Table 4). These effects can be explained by the fact that binding to the biopolymer affects the balance between monomeric and aggregated forms of the dyes (unstructured aggregates that are initially present in aqueous solution). As the concentration of biomolecules increases, the amount and, hence, the fluorescence of aggregated forms decrease, whereas the fluorescence quantum yield of monomeric MCDs bound to dsDNA increases dramatically (Table 4). Depending on the positions and intensities of the fluorescence bands of the initial aggregates, the maxima can shift differently.

In the presence of ssDNA (*c*_ssDNA_ = 0–1.4 × 10^−4^ mol L^−1^), as in the case of dsDNA, the fluorescence excitation spectra of MCD **1**–**4** represent one band corresponding to bound monomeric forms. No new bands appear in the spectra corresponding to the structured H- or J-aggregates of the dyes (Figure 3, Figure 4 and Figure 5; Appendix A in Appendix A).

Despite the fact that for MCD **2** at a low concentration of ssDNA (*c*_rel(ssDNA)_ = *c*_ssDNA_/*c*_MCD_ ≤ 0.14–0.3, hyperchromia of absorption bands is observed (an increase of 21% at *c*_ssDNA_ = 1.7 × 10^−7^ mol L^−1^), a further increase in *c*_rel(ssDNA)_ leads to a decrease in the absorption band intensity of the dye (hypochromia) (Figure 5A). For MCD **1**, **3**, and **4**, the interaction with ssDNA leads to a decrease in the apparent extinction coefficients of the dyes over the entire range of biopolymer concentrations (including low *c*_rel_). In particular, the observed extinction coefficients of MCD **1** and **4** decrease by 40% and 54%, respectively, at *c*_ssDNA_~8 × 10^−5^ mol L^−1^. The hypochromia is accompanied by bathochromic shifts of the maxima and “centers of gravity” of the absorption bands (Δλ_absDNA_ and Δ*M*^−1^ are given in Table 4; the dependence of absorbance and λ_abs_^max^ of MCD **1** on *c*_ssDNA_ is shown in Appendix A of Appendix A).

The influence of ssDNA on the properties of MCD **1**–**4** (spectral shifts of the maxima, positions of the centers of gravity of the absorption bands, their width) was found to be generally comparable with the observed effects of dsDNA. The positions of the band maxima in the fluorescence and fluorescence excitation spectra of MCD in the presence of ssDNA are given in Table 4.

In the presence of HSA, for MCD **1**, in the region of excess dye concentrations (at *c*_rel(HSA)_ ≤ 0.5), the interaction with albumin leads to a rather sharp decrease in the intensity of the dye absorption band. In particular, at *c*_HSA_ = 2.5 × 10^−7^ mol L^−1^, hypochromia is ~25%. A further increase in the albumin concentration is accompanied by a less sharp decrease in the apparent extinction coefficients of the dye (at *c*_HSA_~9 × 10^−6^ mol L^−1^, hypochromia of MCD **1** reaches~36% (*c*_rel(HSA)_ = 3) (Appendix A in Appendix A). A further increase in *c*_rel(HSA)_ does not lead to significant changes in the apparent extinction coefficient. Similarly, for MCD **2** (Appendix A in Appendix A) and **4**, the interaction with HSA leads to a 1.5–2-fold drop in the apparent extinction coefficients of the dyes; for MCD **3**, an even more pronounced effect is observed (Appendix A in Appendix A). The maxima and “centers of gravity” of the absorption spectra undergo small, long-wavelength shifts (Δλ_absHSA_ and Δ*M*^−1^, see Table 4). This indicates that the interaction (binding) with HSA induces aggregation of MCD **1**–**4** (the dyes tend to form unstructured aggregates with low absorption in the bound state).

In the presence of HSA, the fluorescence spectra of MCD **1**–**4** become somewhat narrower and undergo shifts of the maxima (see Table 4). The fluorescence excitation spectra undergo insignificant (~2 nm) long-wavelength shifts and correspond to the monomeric forms of the dyes (Appendix A in Appendix A).

The interaction of MCD **1**–**4** with DNA and HSA leads to a strong increase in the fluorescence intensity of the dyes. This is due to the hindering of nonradiative deactivation processes in dye molecules complexed with biomacromolecules [41]. Plots of the fluorescence intensities of MCD **1**–**4** as functions of dsDNS, ssDNA, and HSA concentrations are shown in Figure 6; such plots at low biopolymer concentrations are presented in Appendix A (Appendix A). In the case of the interaction of MCD **1** with dsDNA (*c*_dsDNA_ = 1 × 10^−4^ mol L^−1^), a relative increase in the fluorescence intensity (*I*_fl_/*I*_fl0_) of about 425 times was observed. For MCD **2**, **3,** and **4** under the same conditions, *I*_fl_/*I*_fl0_ was 173, 79, and 115 times, respectively.

The interaction of MCD with ssDNA leads to a smaller increase in the fluorescence intensity than with dsDNA. For example, for MCD **1** *I*_fl_/*I*_fl0_~40, for MCD **2** *I*_fl_/*I*_fl0_~104 at *c*_ssDNA_ = 5 × 10^−5^ mol L^−1^. In the case of the interaction of MCD with HSA, the increase in the fluorescence quantum yield of the dyes is lower than in the case of dsDNA, with the exception of the fluorescence quantum yield of MCD **3**, which reaches 50% in the presence of 5 × 10^−5^ mol L^−1^ HSA (Table 4). A significant increase in the fluorescence intensity indicates the formation of MCD–HSA complexes in the monomeric (strongly fluorescent) form of the dye.

### 2.3. Molecular Docking of MCD **1**–**4** with dsDNA and HSA

The interaction of MCD with dsDNA can generally proceed with the formation of complexes of various types. Complexes of monomeric molecules of MCD are generally characterized by two main modes of interaction: dyes can intercalate (half or completely) between dsDNA base pairs or bind “on the surface” (in the grooves of the helix) of DNA [41]. According to the half-intercalation model of dye–DNA complex formation, ligand binding to dsDNA is accompanied by intercalation of only one terminal fragment (benzothiazole or benzoxazole terminal nucleus), whereas the second fragment remains exposed in the DNA groove [28,29]. To test the formation of complexes of these two types for MCD **1**–**4**, in silico molecular docking experiments were performed in AMDock 1.5.2 [49]; the results are shown in Table 5.

Docking has confirmed that monomers of MCD **1**–**4** are capable of forming two types of complexes with dsDNA (see Figure 7A,B,E,F). At the same time, binding “on the surface” in the minor groove of the DNA helix was found to be somewhat more favorable for MCD **1**, **3**, and **4**, the free energy of binding ΔG_est_ for this type of complex is lower by 2–2.4 kcal mol^−1^ than for the intercalation complexes (for MCD **2** this difference is less).

Docking of MCD **1** in the minor groove of dsDNA resulted in low distortion, almost planar configurations of the molecule (see Figure 7A), with both heteronuclei and the triethylammonium group deepening into the groove of the biomacromolecule. For MCD **2**, the pyridine group and the quinoline heterocycle are located deep in the groove, while the benzoxazole fragment is located at the outer boundary and is somewhat rotated relative to the plane of the quinoline heterocycle (see Figure 7E). Symmetrical MCD **3** and **4** are characterized by almost planar molecular configurations, with N-substituents pointing outward (see Appendix A in Appendix A).

Almost planar ligand configurations were obtained for the intercalation complexes of MCD **1** and **2** (Figure 7B,F). For MCD **1**, a complex structure close to the half-intercalation type was obtained (the benzothiazole nucleus of the dye is located almost in the groove of the helix; Figure 7B), and the dye is able to form hydrogen bonds (between the -OH group and DNA bases, not indicated in the figure). For MCD **3** and **4**, the simulation showed the half-intercalation type to be preferred, with the geometry of the molecules appearing to be quite “twisted” (see Appendix A in Appendix A).

Docking with HSA showed weaker interactions than in the case of dsDNA (Table 5). The compact size of MCD **1**–**4** molecules allows them to fit into the hydrophobic pockets of HSA (see Figure 7C,D,G,H). The Δ*G*_est_ values for subdomain I were found to be ~0.59–1.35 kcal mol^−1^ higher than for subdomain II. Docking gives even more “twisted” (more distorted) configurations of MCD molecules than in the case of dsDNA (Figure 7D,H). For MCD **3** (subdomain I) and **4** (subdomain II), the molecular structures are close to perplanar (Appendix A in Appendix A).

Docking has shown that the sum of van der Waals, desolvation, and H-bonding energies for the interaction of MCD **1**–**4** with biomacromolecules has a significant superiority over the electrostatic energy.

### 2.4. Thermal Dissociation of dsDNA in the Presence of MCD **1**–**4**

It is known that noncovalent ligand–dsDNA interactions upon the intercalative type of binding lead to additional stabilization of the biopolymer, which is reflected in an increase in the temperature range of thermal dissociation of the DNA double helix. From the data on the thermal dissociation of dsDNA, it is easy to obtain the melting point *T*_m_(dsDNA) (at which 50% dsDNA is denatured) [51,52]. Note that *T*_m_(dsDNA) depends on its GC composition, molecular length, and degree of strand complementarity [43]; long duplexes with fewer base mismatches will have higher *T*_m_(dsDNA) than short duplexes with more mismatches.

Melting of the DNA double helix was measured spectrophotometrically at *t* = 40–98 °C, the normalized temperature dependences of the absorbances of the samples at λ_abs_ = 260 nm are shown in Figure 8.

Experiments on thermal dissociation (“melting”) of dsDNA (*c*_dsDNA_~6 × 10^−5^ mol L^−1^) in the presence of MCD (*c*_MCD_~6 × 10^−6^ mol L^−1^) showed an increase in *T*_m_(dsDNA) in the presence of MCD, which confirmed the formation of intercalative complexes. In particular, in the presence of MCD **1** and **2**, *T*_m_(dsDNA) = 85–86 °C (in the absence of cyanines, *T*_m_(dsDNA) = 80 °C). For symmetrical MCD **3** and **4**, *T*_m_(dsDNA) was obtained to be 82 °C and 83 °C, respectively (the growth of *T*_m_(dsDNA) about 2–3 °C). This indicates a more significant contribution of the intercalative complexes for unsymmetrical MCD **1** and **2** than for symmetrical dyes **3** and **4**.

### 2.5. Study of the Interaction of MCD **1**–**4** with HSA by the Method of Synchronous Fluorescence Scanning

In order to determine the binding sites of HSA when interacting with MCD and to evaluate possible changes in these sites, the fluorescence spectra of tyrosine and tryptophan groups of HSA in the presence of MCD were measured in the synchronous fluorescence scanning (SFS) mode [53]. The positions of the SF spectra of protein amino acid residues depend on their microsolvation (micropolarity of the medium) [54,55]. Thus, SFS allows for obtaining data on the effect of complexation on the microenvironment of fluorophore groups of protein molecules.

An increase in the concentration of MCD **1**–**4** (*c*_MCD_~0–5.3 × 10^−5^ mol L^−1^) does not cause noticeable shifts in the maxima of the SFS bands: for tyrosine groups (Δλ = 15 nm), λ_SFmax_ = 284 nm, for tryptophan residues (Δλ = 60 nm), λ_SFmax_ = 278 nm (Figure 9). It can be concluded that the complexation of MCD **1**–**4** with HSA does not change the microenvironment (or configuration) of the fluorophore groups at the HSA binding sites. At the same time, fluorescent titration of the HSA solution with the dyes showed a significant decrease in the intensity of the SFS bands of both tyrosine and tryptophan. For MCD **1** and **2**, the intensity of the SFS bands for tyrosine decreased by 31% and 36%; for tryptophan, 26% and 20%, respectively. For MCD **4**, a comparable effect was observed (21% and 22% for tyrosine and tryptophan, respectively), and the smallest effect was observed for MCD **3** (11% and 6%, respectively).

Quenching of intrinsic HSA fluorescence was analyzed using the Stern–Volmer and Lehrer–Leavis relationships (see Equations (3) and (4)) [56,57]. The experimental dependences approximated by linear anamorphoses (R^2^~0.9–0.98) were better in the coordinates of the Lehrer–Leavis model than in the Stern–Volmer model (the Stern–Volmer plots were nonlinear). The Lehrer–Leavis model assumes partial accessibility of HSA chromophores for quenching by MCD (the resulting quenching constants are shown in Table 6).

Quenching of intrinsic fluorescence of HSA can occur, in particular, due to energy transfer (Förster resonance energy transfer, FRET) from protein donor fluorophores to MCD acceptor molecules. In the MCD–HSA system, complexation brings acceptors and donors closer together and facilitates FRET. FRET is likely because HSA fluorescence and MCD absorption spectra overlap. For MCD and HSA, the overlap integrals *J* (see Equation (S7) in Appendix A) were obtained in the order of 2.0–7.0 × 10^15^ mol^−1^ cm^−1^ nm^4^ (for MCD **4** see Appendix A in Appendix A). The Förster critical radius *R*_0_ (the distance with a 50% probability of HSA donor fluorophore quenching by MCD acceptors; see Equation (S8) in Appendix A) was found to be 40–45 Å. For the shorter-wavelength absorbing MCD **3**, *R*_0_~50 Å, and the highest value of *J* was obtained, which is in contradiction with the low degrees of quenching of the SFS bands by MCD **3** mentioned above. This is probably due to the strong aggregation of MCD **3** in the presence of HSA, which decreases the concentration of monomeric dye molecules active in fluorescence quenching (see Appendix A, Appendix A). Relatively high values of *K*_SV_ for MCD **3** presented in Table 6 were obtained at very low dye concentrations (at the initial portion of the quenching plots) when dye aggregation is still insignificant. But as the dye concentration increases, its aggregation is built up steeply, and the fluorescence quenching diminishes.

Since the fluorescence decay time for HSA τ_0_~10^−8^ s [58], the values of the dynamic quenching rate constants *k*_q_ (*k*_q_ = *K*_SV_/τ_0_) for MCD **1**–**4** are in the range of 2–4 × 10^12^ L mol^−1^ s^−1^, which is much higher than the diffusion limit (in aqueous solutions, *k*_diff_~10^10^ L mol^−1^ s^−1^). This indicates that the observed effects of quenching of HSA intrinsic fluorescence are static in nature, that is, quenching occurs in the MCD–HSA complex. The quenching process is not limited by the diffusion of the quencher to the fluorophore centers of the protein in solution but is determined by the features of complexation.

### 2.6. Binding Constants, Limits of Detection/Quantification

The complication of chemical equilibrium, and the variety of interactions between MCD and biopolymer (dye aggregation/deaggregation, the possibility of forming complexes of several types with DNA/HSA) makes it problematic to determine the exact values of the complexation constants of the dyes with DNA. The values of the effective equilibrium constants of the complexation reaction of MCD with biopolymers (*K*_eff_, L mol^−1^, see Table 7) were estimated from spectral fluorescent data. In this case, we had to assume a simple equilibrium MCD (monomer, free) ⇄ MCD (monomer, bound). For these purposes, we used the Benesi–Hildebrand [59], Hill [60], and Scatchard [61] formalisms (See Appendix A, Equations (S9)–(S13)). For dsDNA, fitting according to the modified Scatchard equation was carried out assuming one dye molecule per one DNA turn (*K*_1Sc_ at *n*_Sc_ = const = 10 b.p.), as well as varying both the constant (*K*_2Sc_) and the stoichiometric coefficient (*n*_Sc_). In the case of ssDNA, *n*_Sc_ is given per one base; for HSA, *n*_Sc_ = 1.

For the interaction of MCD **1** with dsDNA, the highest values of effective complexation constants were obtained (*K*_effBH_, *K*_effHill_, *K*_1Sc_, *K*_2Sc_). At c_dsDNA_ = 5 × 10^−5^ mol L^−1^, almost 80% of the dye is complexed and gives the highest fluorescent response. For MCD **2**–**4**, smaller values were obtained. Insignificant deviations of *n*_Hill_ from unity for MCD **1** and **2** with dsDNA and MCD **2** with ssDNA and HSA indicate the absence of cooperative binding effects; in other cases, *n*_Hill_ is ~0.57–0.89, which indicates anticooperative effects of complex formation (binding of one MCD molecule into a complex hinders the interaction of others).

Note that in the case of ssDNA and HSA, the complexation constants were found to be several times lower than for dsDNA. Similar effects were observed earlier for other MCDs: in particular, it was shown that the interaction with ssDNA of SYBR Green I led to a significantly lower fluorescence increase (about 11 times lower than with dsDNA) [62]. It is intercalation that is the main type of binding to dsDNA for the SYBR Green I analog—monomethine cyanine PicoGreen [63]. The experiments on thermal dissociation of dsDNA in the presence of MCD **1**–**4** also confirm a significant contribution of intercalation to the binding of the dyes to dsDNA. Thus, the reason for lower *K*_eff_ and the smaller fraction of bound dye (α_b_) for MCD **1**–**4** with ssDNA is probably the lack of the possibility of intercalative interaction.

As to HSA, it is known that the formation of complexes with serum albumins generally proceeds more efficiently for anionic cyanine dyes than for cationic ones [64,65,66,67]. This may explain lower *K*_eff_ values obtained for complexation of cationic MCD **1** and **4** with HSA than with dsDNA. However, for MCD **3** and, particularly, **2** the *K*_eff_ values for HSA are higher than those for dsDNA. It is known that some cationic MCDs interact effectively (with a 100-fold increase in fluorescence intensity) with serum albumin [68]. The molecules of those dyes have additional aromatic cyclic ‘‘affinity-modifying groups’’ linked to chromophores by flexible linkers. Possibly, the pyridinium group in the molecule of MCD **2**, bound to the chromophore by a flexible hexamethylene chain, plays the role of such “affinity-modifying group” and is responsible for increasing the *K*_eff_ value for this dye with HSA.

It should also be noted that high binding constants with HSA for MCD **2** and **3** were obtained at the initial stage of fluorescence growth (at low HSA concentrations of about 5 × 10^−6^ mol L^−1^ or less). With a further increase in the HSA concentration, the dependence of the fluorescence intensity on the HSA concentration rapidly leveled off at a value of about three times lower than in the case of dsDNA and may be due to the strong aggregation of the dyes on HSA. This leads to a smaller increase in the fluorescence growth for MCD in the presence of HSA (at high concentrations) than in the presence of DNA.

To characterize MCD **1**–**4** as possible fluorescent probes for dsDNA, the values of LOD and LOQ for different MCD have been determined. We have found that MCD **1** has the best characteristics—the lowest LOD and LOQ values (about 10^−8^ mol L^−1^, with a linear concentration interval of 0–3 µmol L^−1^; see Table 7). Taking into account its much lower sensitivity to HSA (about 37 times), MCD **1** can serve as a sensitive and selective probe for dsDNA in the presence of HSA. Slightly less sensitive to dsDNA is MCD **2**, but it does not have such selectivity. At the same time, by its sensitivity, it could serve as a fluorescent probe for HSA (LOD = 3.43 × 10^−8^ mol L^−1^).

### 2.7. Photochemical Properties of MCD **1**–**4** in the Presence of DNA

Upon direct flash photoexcitation of deoxygenated solutions of MCD **1**–**4**, signals corresponding to triplet–triplet (T–T) absorption were not observed due to the low efficiency of intersystem crossing to the triplet (T) state, in comparison with competing processes of nonradiative deactivation of excited states of dye molecules, which is common for MCD [41]. Complexation with dsDNA hinders these competing processes, which leads to an increase in the quantum yield of the T state of MCD. In the presence of dsDNA, under direct photoexcitation, the signals of T states of MCD **1**–**3** were detected, and their different T–T absorption spectra were measured (Figure 10).

The signal of the T state of MCD **4** upon direct photoexcitation in the presence of dsDNA was not detected. This could be due to its short lifetime and/or low quantum yield, or insufficient sensitivity of the flash photolysis setup (the T quantum yield sensitivity of the experimental setup was about 0.15% [69]). The T state of MCD **4** in the presence of dsDNA was obtained in an experiment on T–T energy transfer using sodium naphthalene-2-sulfonate sensitizer. The corresponding T–T absorption spectrum is shown in Figure 10 (spectrum *4*).

Quantum yields of intersystem crossing to the triplet state (Φ_ST_) of MCD **1**–**3** in the presence of dsDNA were estimated by flash photolysis using optical actinometry (see Materials and Methods). The obtained values of Φ_ST_ for MCD **1**–**3** were 2.6 ± 0.4%, 5.7 ± 0.8%, and 10.7 ± 1.5%, respectively.

In the presence of dsDNA, the lifetimes of the T state (τ_T_) of MCD **1**–**3** were found to be 0.12–0.22 ms; for MCD **4**, τ_T_~0.03 ms. The decay kinetics of the T state of MCD **1**–**3** had a two-exponential character: [^3^Dye*] = *A*_1_exp(–*k*_1_*t*) + *A*_2_exp(–*k*_2_*t*). In particular, for MCD **1**, the decay rate constants of the T state were ~9.9 × 10^3^ and 2 × 10^3^ s^−1^; for MCD **3**, *k*_1_~1.3 × 10^4^ s^−1^ and *k*_2_~4 × 10^3^ s^−1^. This can be explained by the formation of two types of noncovalent complexes with the biomacromolecule (binding in the dsDNA groove and the intercalation complex). In this case, the microenvironment of ligand molecules will be different, and the triplet states of dye molecules will have different kinetic characteristics.

Quenching of the triplet state of the dyes in the presence of dsDNA by dissolved oxygen was studied (the quenching rate constants were determined from the linear dependences of the T state decay rate constants on the oxygen concentration, see Appendix A in Appendix A). As for the previously studied thia- and oxacarbocyanine dyes in the presence of DNA [46,47,48], the quenching rate constant (*k*_qO2_) of the main (short-lived) component of the triplet state was significantly lower than the values for diffusion-controlled reactions. In particular, for MCD **2** *k*q_O2_ = 1.7 × 10^8^ mol^−1^ L s^−1^ and for MCD **3** kq_O2_ = 3.6 × 10^8^ mol^−1^ L s^−1^). For aqueous solutions, taking into account the spin-statistical factor of the interaction of a triplet molecule with triplet molecular oxygen (1/9), *k*_qO2_ should be equal to 1/9 *k*_dif_~3 × 10^9^ mol^−1^ L s^−1^. This decrease is because of the fact that the complexation of polymethine dyes with biomacromolecules probably leads to partial shielding of dye ligands, which leads to hindered access of quencher molecules (in particular, dissolved oxygen) to the T state of the dyes and a decrease in the quenching rate constant [70,71,72].

The increase in the quantum yield of the T state of MCD in complexes with biomolecules may be of importance for using these dyes as potential candidates in photodynamic therapy (similarly to MCD synthesized earlier [73]). In the case of MCD **1**–**3**, triplet dye molecules will be formed only in a complex with a biomolecule, which imparts to these dyes targeted characteristics when used in photodynamic therapy.

## 3. Materials and Methods

MCD **1** and **2** were obtained from the Institute of Molecular Biology and Genetics of the National Academy of Sciences of Ukraine (courtesy of Prof. S.M. Yarmoluk); MCD **3** and **4** were obtained from N.D. Zelinsky Institute of Organic Chemistry, Russian Academy of Sciences (provided by Prof. Zh.A. Krasnaya).

Distilled water, 0.01 M aqueous phosphate buffer saline solution (PBS; pH 7.4, containing 0.137 M NaCl and 0.0027 M KCl), ethanol, benzyl alcohol, isopentyl alcohol, acetonitrile, DMSO, 1,4-dioxane, chloroform, DMSO, acetone, methyl ethyl ketone, ethyl acetate (all analytical grade). Commercial chick erythrocyte DNA (Reanal, Budapest, Hungary) and human serum albumin (HSA, Sigma-Aldrich, St. Louis, MO, USA) were used.

Single-stranded form of DNA (single-stranded DNA; ssDNA) was obtained according to the method described in [74]. The concentrations of DNA and the dyes were controlled spectrophotometrically. Molar extinction coefficient of dsDNA (ε, L⋅mol^−1^⋅cm^−1^) was taken to be 13,200 L mol^−1^ cm^−1^ (per base pair) [75]; for ssDNA, ε = 9045 L mol^−1^ cm^−1^ (per one base) [76] was used. For MCD **1** and **2** in an aqueous buffer solution, we used extinction coefficients ε = 40,000 and 68,000 L mol^−1^ cm^−1^, respectively, which were taken from [26]. For MCD **3** and **4**, the molar extinction coefficients in ethanol solution were 69,200 and 49,000 L mol^−1^ cm^−1^ (taken from [42]). The extinction coefficients in buffer solutions **3** and **4** were found to be 61,800 and 44,500 L mol^−1^ cm^−1^, respectively [42]. Spectral fluorescent characteristics of MCD in complexes with biomacromolecules were studied by titration of dye solutions (see Appendix A). The thermal dissociation (“melting”) of dsDNA in the presence of MCD was studied spectrophotometrically (λ_abs_ = 260 nm), the sample was heated (*t*° = 30–95 °C) using Peltier cells, and the temperature was controlled using a thermocouple. Melting points *t*_m_, corresponding to 50% hyperchromia of DNA samples were obtained [51,52,77].

Absorption spectra were recorded on an SF-2000 spectrophotometer (Experimental-Construction Company “Spectrum”, St. Petersburg, Russia); fluorescence and fluorescence excitation spectra were recorded on a Fluorat-02-Panorama spectrofluorimeter (Group of Companies “Lumex”, St. Petersburg, Russia). The fluorescence and fluorescence excitation spectra were corrected for the spectral characteristics of the instrument (excitation channel signal) and sample transmission. Changes in the intrinsic fluorescence of HSA upon binding of MCD (in the presence of the dyes) were recorded in the synchronous fluorescence scanning (SFS) mode. In SFS mode, when registering spectra, a fixed difference Δλ= λ_fl_ − λ_ex_ is maintained for the channels of excitation and registration [53,54,55,78]. Synchronous scanning spectra were obtained at Δλ = 15 nm (fluorescence of tyrosine residues in the HSA molecule) and Δλ = 60 nm (fluorescence of tryptophan) [53,78,79]. The spectra were measured in 1 cm quartz semi-micro cells (Bio-Rad, Hercules, CA, USA).

The fluorescence quantum yields of MCD **1**–**4** (φ_fl_) were determined by comparison with a standard (see Appendix A, Equation (S1)) [56]. Fluorescein was used as a standard: Φ_fl_._st._ = 0.93% in 0.1 M NaOH (aq) [80]. The dye solutions with maximum absorbances of about 0.1 were used. Since the refractive index of 0.1 M NaOH aqueous solution differs very little from that of water [81], the refractive index of water was used for the standard solution.

The rate constants of radiative and nonradiative deactivation (*k_r_* and *k_nr_*, respectively), for MCD were estimated from Equations (1) and (2) [82]:(1)kr=1τr=2900n2ν˜02∫ε dν˜
and
(2)$$k_{nr}=k_{r}\frac{1-\varphi_{fl}}{\varphi_{fl}}$$
where *τ_r_* is the radiative lifetime, ${\tilde{\nu }}_{0}^{}$ is the dye absorption maximum (in terms of wavenumbers expressed in μm^−1^), *n* is the refractive index of the medium, and $\int \varepsilon \;d\tilde{\nu }$ is the integral of the absorption spectral curve.

The Stern–Volmer (Equation (3)) and Lehrer–Leavis (Equation (4)) dependences were used to analyze the results of measurements of HSA intrinsic fluorescence (synchronous scanning mode) in the presence of MCD [56,57].
(3)$$I_{0}/I=1\;+\;K_{SV}[Q]$$
(4)$$I_{0}/(I_{0}-I)=1\;/\alpha +\;1/(\alpha K_{SV}[Q])$$
where *I*_0_ and *I* are the fluorescence intensities in the absence and in the presence of a quencher, respectively; [*Q*] is the quencher concentration; *K_SV_* is the Stern–Volmer quenching constant; and *α* is the proportion of fluorophores in the biomolecule available to the quencher in the Lehrer–Leavis model.

The spectral–kinetic characteristics of the T state of MCD were studied by flash photolysis. The experiment was carried out on a setup with excitation by a xenon lamp (lamp flash energy ~50 J, duration τ_1/2_ = 10 μs). When registering T states, the solutions were deoxygenated using a vacuum setup. The manometric unit was used to determine the air pressure in a series of experiments on T state quenching by dissolved oxygen. The T state of MCD **4** in PBS solution in the presence of dsDNA was obtained by energy transfer from the triplet level of sodium naphthalene-2-sulfonate (*E*_T_~21,000 cm^−1^ [83]). The quantum yields of T states (Φ_T_) of the dyes in the presence of dsDNA were estimated actinometrically. Dyes with known Φ_T_ were used as actinometers: the xanthene dye Acid Red 87 (Φ_T_ = 0.8 (aq); [84]) and the cyanine dye 3,3′-diethyl-9-methoxythiacarbocyanine iodide (Φ_T_ = 0.16–0.20 in n-propanol; [85]).

Spectral measurements, except for DNA melting experiments, were carried out at room temperature (23 ± 2 °C).

The evaluation of the polar, hydrophobic/lipophilic properties of MCD was given in terms of the polar surface area of dye molecules (TPSA, Å^2^) and dimensionless distribution coefficients in the octanol–water system (logP) using the www.molinspiration.com service [48] (accessed on 2 August 2023).

The method of spectral moments has been used for the analysis of the dye absorption spectra to more accurately describe changes in the spectra (see Supplementary Material, Equations (S2)–(S6)) [46].

The parameters of FRET between albumin donor chromophores (tyrosine and tryptophan amino acid residues of HSA) and acceptors (MCD **1**–**4** molecules) were determined from the spectral fluorescent data using the standard formulae for the spectral overlap integral *J* (M^−1^ cm^−1^ nm^4^) and critical radius of energy transfer *R*_0_ (see Supplementary Material, Equations (S7) and (S8), respectively) [56].

The equilibrium constants of MCD complexation with biomolecules (*K*_a_, L mol^−1^) were obtained from fluorescence spectra using the Benesi–Hildebrand [59], Hill [60], and Scatchard equations [61] (see Appendix A, Equations (S9)–(S13)). Limits of detection (LOD) and quantification (LOQ) of biomolecules were determined from concentration plots of MCD fluorescence intensities [86] (see Appendix A, Equations (S14) and (S15)).

Modeling and analysis of the structure of dye complexes with DNA and HSA was performed by molecular docking using AMDock 1.5.2 [49]. UCSF Chimera [50] and Avogadro 1.2.0 [87] were used to prepare the MCD and DNA structures for docking (adding hydrogen atoms, optimizing the ligand structure) and visualizing the results. The PDB 1BNA structure of B-DNA (complex in the DNA groove, [88]) and 198D (modeling of the intercalation complex [89]). The HSA structure was also obtained from the Protein Data Bank (4K2C; [90]). For 1BNA, the grid was 37 × 37 × 40 (center x = 15, y = 20, z = 10), for 198D the grid was 20 × 25 × 25 (center x = 15, y = 25, z = 10). For HSA, a 24 × 24 × 24 Å grid was used; the grid centers corresponded to Sudlow binding sites I (residue TRP214; x = 4.3, y = –0.5, z = –6) and II (TYR411; x = 24.6, y = –5.05, z = 8.0).

## 4. Conclusions

In search of new probes for biomolecules, the spectral fluorescent study of MCD **1**–**4**, both unsymmetrical and symmetrical, has been carried out in different organic solvents, in aqueous buffer solutions, and in the presence of biomacromolecules—DNA and HSA. There were no distinct correlations found between the absorption maxima of the dyes and the polarity/polarizability of the solvents, which suggests that the electronic distribution in the dye molecule is practically unaffected by the solvent polarity/polarizability. In the presence of DNA, the absorption spectra of MCD **1**–**4** show mostly a decrease in the apparent extinction coefficients (hypochromia) and slight bathochromic shifts of the absorption bands, which reflects the spectral characteristics of the dye forms complexed with DNA. Hypochromia of the dyes is also observed in the presence of HSA. The interaction of MCD **1**–**4** with DNA and HSA leads to a steep growth of the fluorescence intensity due to the hindering of nonradiative deactivation in dye molecules complexed with biomacromolecules. Complexes of MCD with dsDNA and HSA were modeled in silico by molecular docking. It has been shown that, with dsDNA, both minor groove and intercalation (half-intercalation) complexes are formed; with HSA, complexes in Sudlow’s sites I and II are possible. Experiments on thermal dissociation (“melting”) of dsDNA in the presence of MCD showed the contribution of intercalative complexes, more significant for unsymmetrical dyes **1** and **2** than for symmetrical dyes **3** and **4**. Quenching of intrinsic fluorescence of HSA by MCD **1**–**4** (using synchronous fluorescence scanning) occurs with rate constants much higher than the diffusion limit, which indicates the formation of the dye–HSA complex. Effective constants of complexation of MCD **1**–**4** with the biomacromolecules were estimated using Benesi–Hildebrand, Hill, and Scatchard formalisms. MCD **1** has the best characteristics as possible fluorescent probe for dsDNA—the lowest LOD and LOQ values (about 10^−8^ mol L^−1^). Taking into account its much lower sensitivity to HSA, MCD **1** can serve as a sensitive and selective probe for dsDNA in the presence of HSA.

Upon direct flash photoexcitation of deoxygenated solutions of MCD **1**–**3** in the presence of dsDNA, signals corresponding to triplet–triplet absorption were observed, which were absent without DNA. The increase in the quantum yield of the triplet state of MCD in complexes with biomolecules may be of importance for using these dyes as potential candidates in photodynamic therapy.

## Figures and Tables

**Figure 1 ijms-24-13954-f001:**
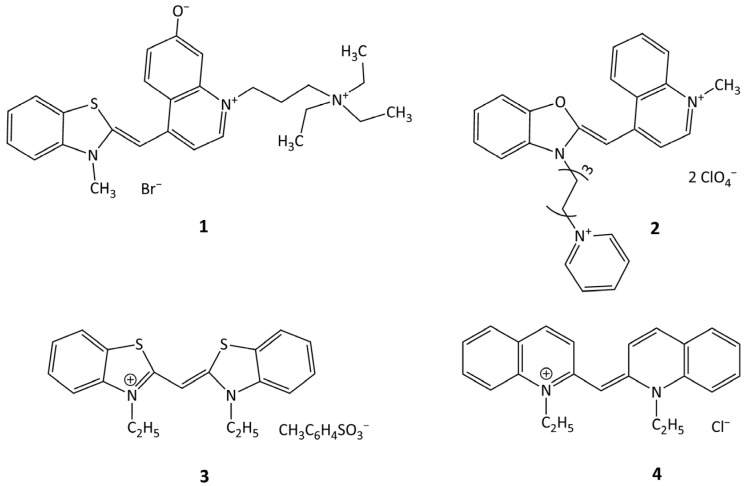
Structures of monomethine cyanine dyes **1**–**4**.

**Figure 2 ijms-24-13954-f002:**
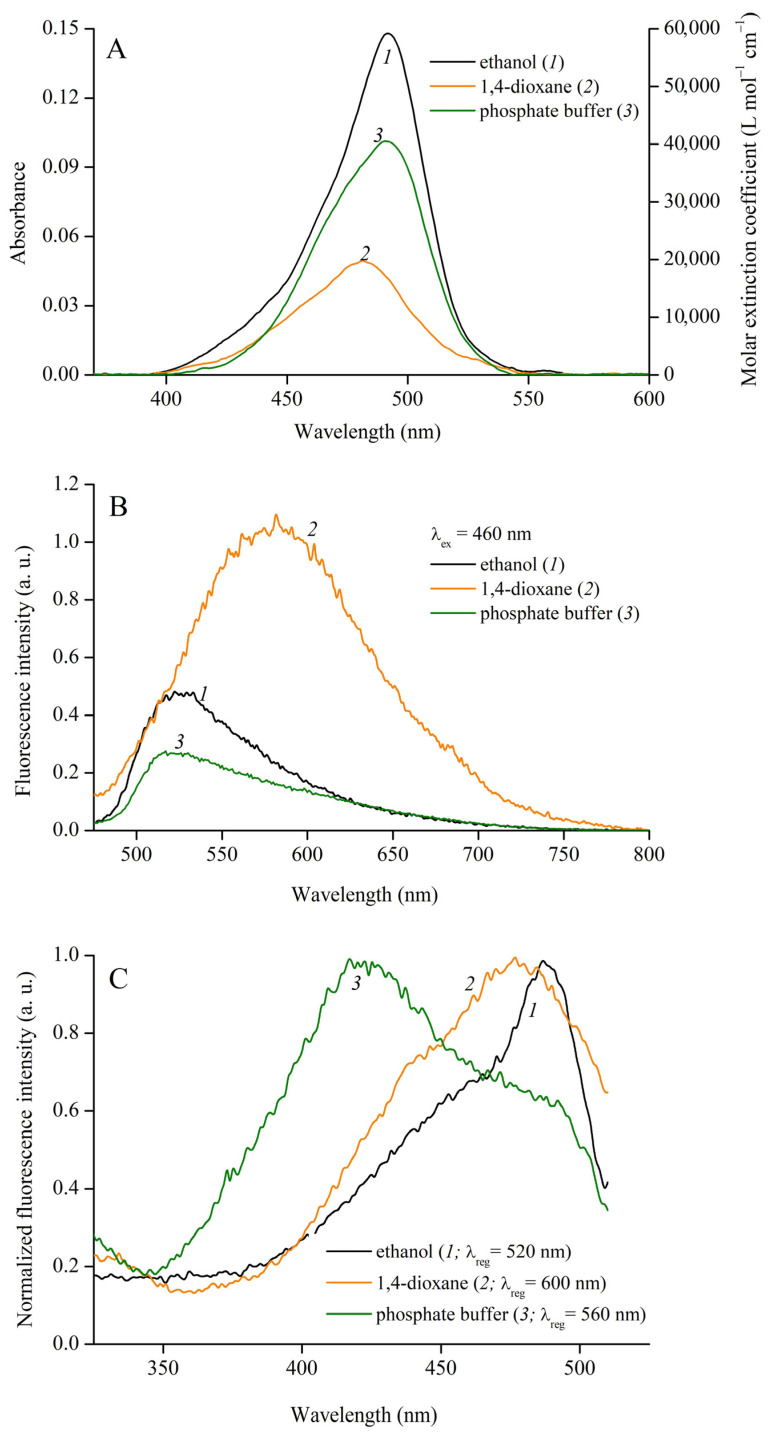
Absorption (**A**), fluorescence ((**B**), λ_ex_ = 460 nm), and normalized fluorescence excitation spectra (**C**) of MCD **1** (*c*_MCD**1**_ = 2.6 × 10^−6^ mol L^−1^) in ethanol (*1*), 1,4-dioxane (*2*), and phosphate buffer (PBS) pH 7.4 (*3*) solutions. The fluorescence excitation spectra were registered at λ_reg_ = 520 (*1*), 600 (*2*), and 560 (*3*) nm. Molar extinction coefficients (ε, L mol^−1^ cm^−1^) of MCD **1** are given (a, right-hand axis).

**Figure 3 ijms-24-13954-f003:**
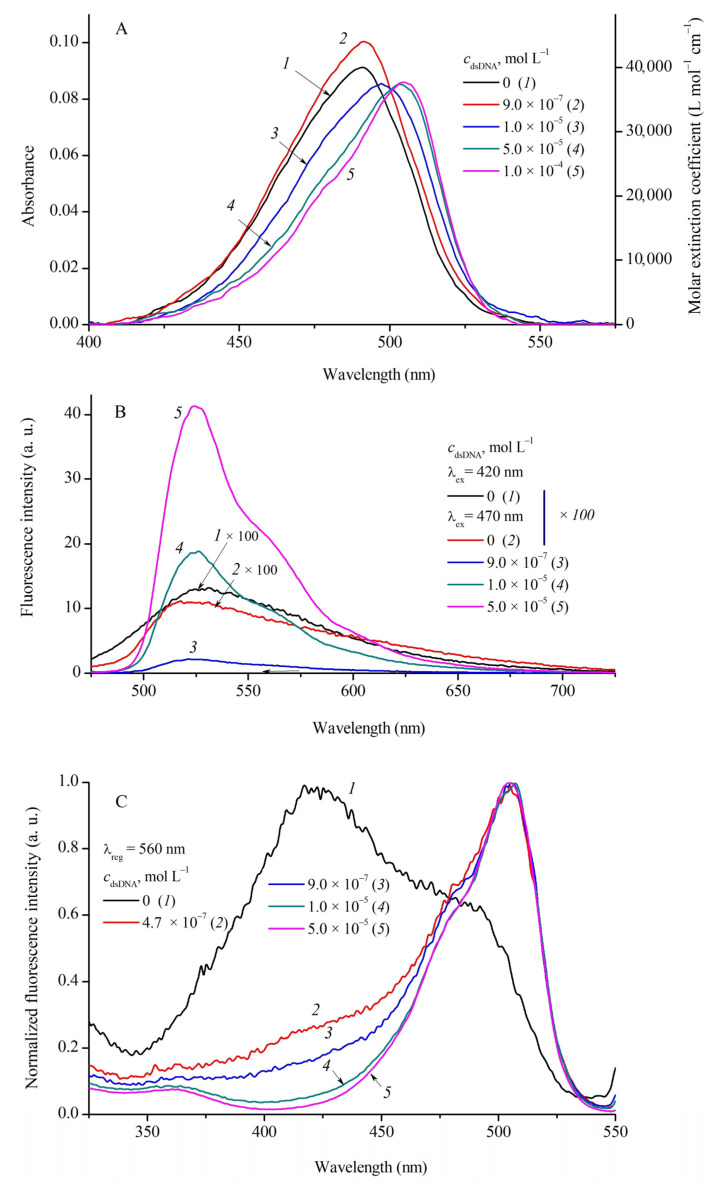
Absorption (**A**), fluorescence ((**B**); *1b* was recorded at λ_ex_ = 420 nm; spectra *2b–5b* were recorded at λ_ex_ = 470 nm), and normalized fluorescence excitation spectra ((**C**); λ_reg_ = 560 nm) of MCD **1** (*c*_MCD**1**_ = 2.3 × 10^−6^ mol L^−1^) in the presence of dsDNA: 0 (curves *1*a*–1*c**, *2b*), 4.7 × 10^−7^ (*2c*), 9.0 × 10^−7^ (*2*a**, *3b*, *3*c**), 1.0 × 10^−5^ (*3*a**, *4b*, *4*c**), 5 × 10^−5^ (*4*a**, *5b*, *5*c**), and 1 × 10^−4^ (*5*a**) mol L^−1^ dsDNA. Molar extinction coefficients (ε, L mol^−1^ cm^−1^) of MCD **1** are given (a, right-hand axis).

**Figure 4 ijms-24-13954-f004:**
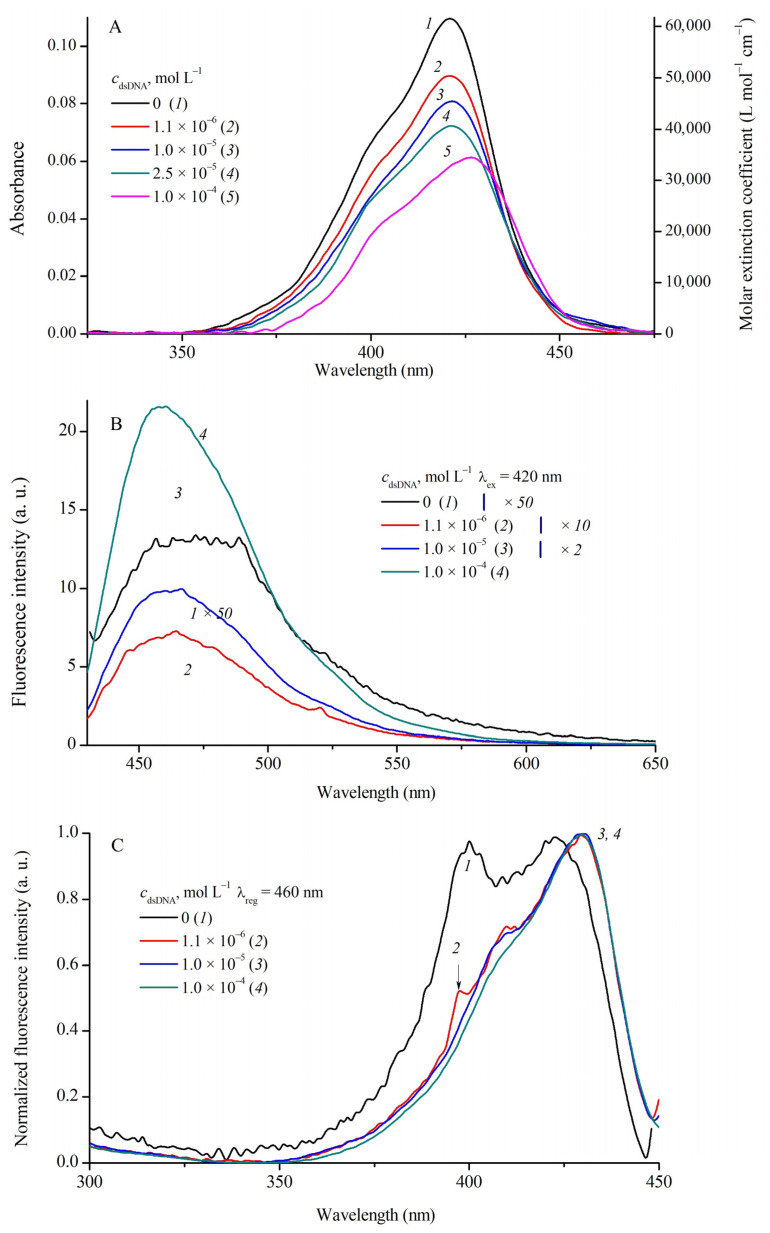
Absorption (**A**), fluorescence ((**B**); λ_ex_ = 420 nm), and normalized fluorescence excitation spectra ((**C**); λ_reg_ = 460 nm) of MCD **3** (*c*_MCD**3**_ = 1.7 × 10^−6^ mol L^−1^) in the presence of dsDNA: 0 (curves *1*a**–*1*c**), 1.1 × 10^−6^ (*2*a**–*2*c**), 1.0 × 10^−5^ (*3*a**–*3*c**), 2.5 × 10^−5^ (*4*a**), and 1 × 10^−4^ (*5*a**, *4b*, *4*c**) mol L^−1^ dsDNA. Molar extinction coefficients (ε, L mol^−1^ cm^−1^) of MCD **3** are given (a, right-hand axis).

**Figure 5 ijms-24-13954-f005:**
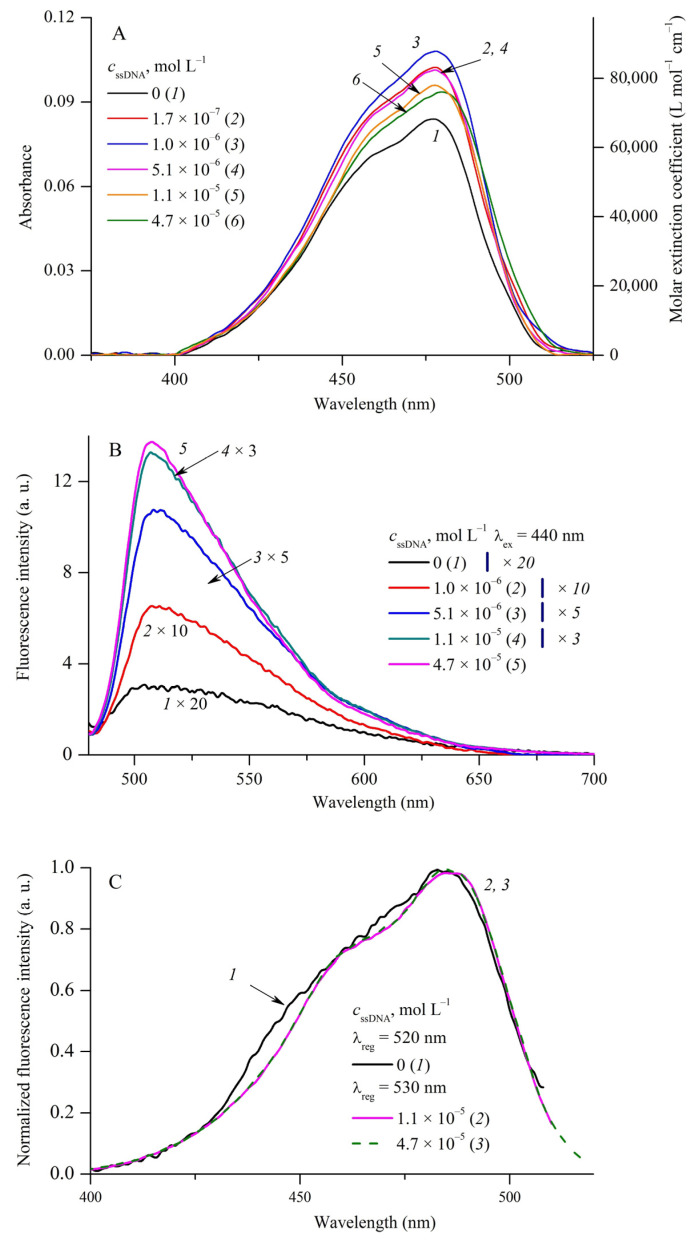
Absorption (**A**), fluorescence ((**B**); λ_ex_ = 470 nm), and normalized fluorescence excitation spectra ((**C**); *1c* was recorded at λ_reg_ = 520 nm; spectra *2c, 3c* were recorded at λ_reg_ = 530 nm) of MCD **2** (*c*_MCD**2**_ = 1.2 × 10^−6^ mol L^−1^) in the presence of ssDNA: 0 (curves *1*a**–*1*c**), 1.7 × 10^−7^ (*2a*), 1.0 × 10^−6^ (*3*a**, *2b*), 5.1 × 10^−6^ (*4*a**, *3b*), 1.1 × 10^−5^ (*5*a**, *4b*, *2*c**), 4.7 × 10^−5^ (*6*a**, *5b*, *3*c**) mol L^−1^ ssDNA. Molar extinction coefficients (ε, L mol^−1^ cm^−1^) of MCD **2** are given (a, right-hand axis).

**Figure 6 ijms-24-13954-f006:**
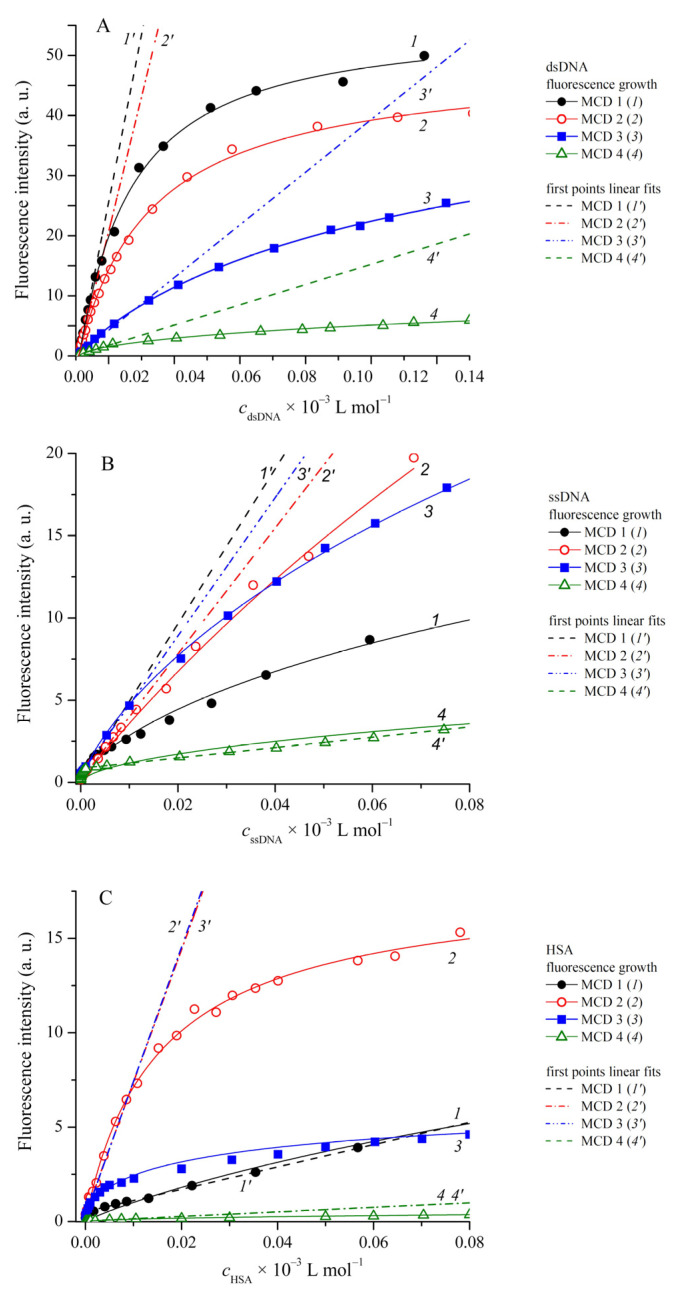
Plots of MCD **1**–**4** fluorescence intensity (*1*–*4*, respectively), and linear fits of their initial portions (*1*′–*4*′) vs. dsDNA (**A**); *c*_MCD**1**_ = 2.3 × 10^−6^, *c*_MCD**2**_ = 1.2 × 10^−6^, *c*_MCD**3**_ = 1.7 × 10^−6^, *c*_MCD**4**_ = 2.6 × 10^−6^ mol L^−1^), ssDNA (**B**); *c*_MCD**1**_ = 2.3 × 10^−6^, *c*_MCD**2**_ = 1.2 × 10^−6^, *c*_MCD**3**_ = 1.0 × 10^−6^, *c*_MCD4_ = 3.2 × 10^−6^ mol L^−1^) and HSA (**C**); *c*_MCD**1**_ = 3.0 × 10^−6^, *c*_MCD**2**_ = 2.4 × 10^−6^, *c*_MCD**3**_ = 1.0 × 10^−6^, *c*_MCD**4**_ = 1.0 × 10^−6^ mol L^−1^) concentrations in PBS buffer.

**Figure 7 ijms-24-13954-f007:**
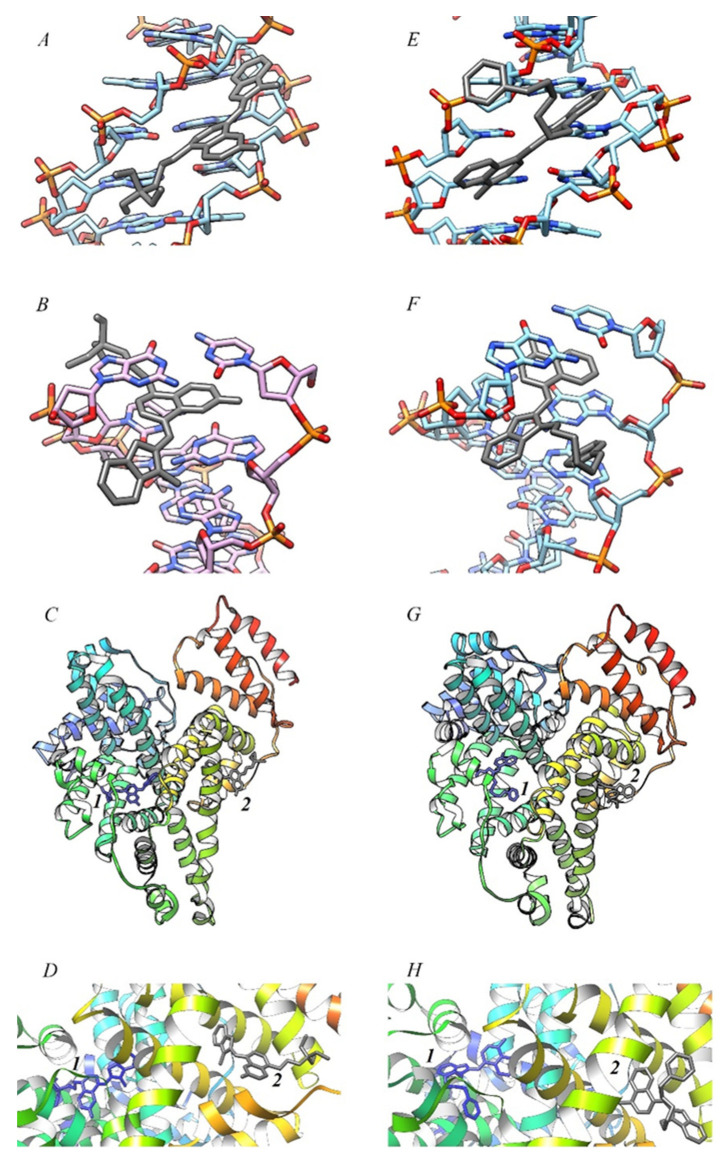
Results of molecular docking in AMDock 1.5.2 of MCD **1** (**A**–**D**) and **2** (**E**–**H**) with dsDNA: complexes in the minor groove (1BNA, (**A**,**E**)), and intercalation (198D; (**B**,**F**)); and with HSA (4K2C): Sudlow’s site I (blue; *1c*, *1d*, *1g*, *1h*) and site II (gray; *2c*, *2d*, *2g*, *2h*). Molecular graphics was performed with UCSF Chimera [50].

**Figure 8 ijms-24-13954-f008:**
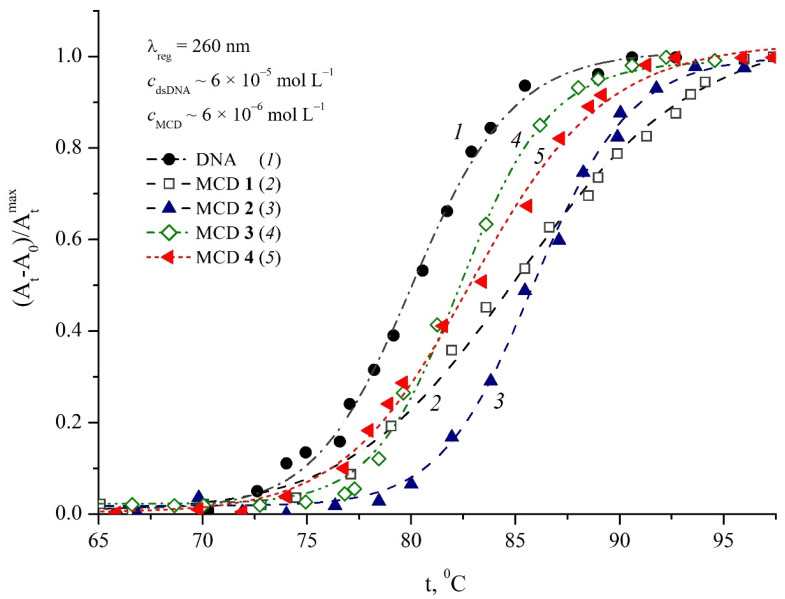
Normalized ((*A*_t_ − *A*_0_)/*A*_tmax_) temperature dependences of thermal dissociation of dsDNA in PBS buffer (λ_reg_ = 260 nm) in the absence of MCD (*1*) and in the presence of MCD **1** (*2*), **2** (*3*), **3** (*4*), and **4** (*5*); *c*_dsDNA_ ~ 6 × 10^−5^ mol L^−1^, *c*_MCD_ ~ 6 × 10^−6^ mol L^−1^.

**Figure 9 ijms-24-13954-f009:**
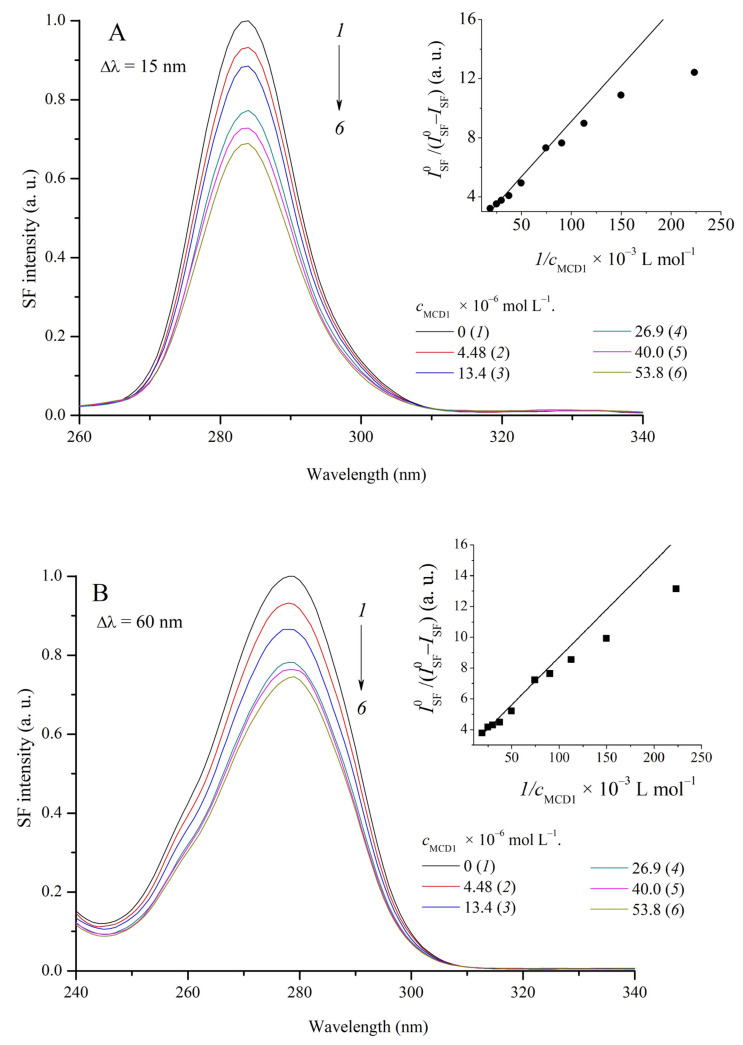
Synchronous fluorescence spectra of HSA (*c*_HSA_ = 1 × 10^−5^ mol L^−1^) at different concentrations of MCD **1**: c_MCD_ = 0 (*1*), 4.48 (*2*), 13.4 (*3*), 26.9 (*4*), 40.03 (*5*), and 53.8 (*6*) × 10^−6^ mol L^−1^. The spectra were recorded with Δλ = 15 (**A**) and 60 (**B**) nm, plotted vs. excitation wavelength. Insets: Lehrer–Leavis plots for quenching of HSA fluorescence, measured using SFS at Δλ = 15 (**A**) and 60 (**B**) nm.

**Figure 10 ijms-24-13954-f010:**
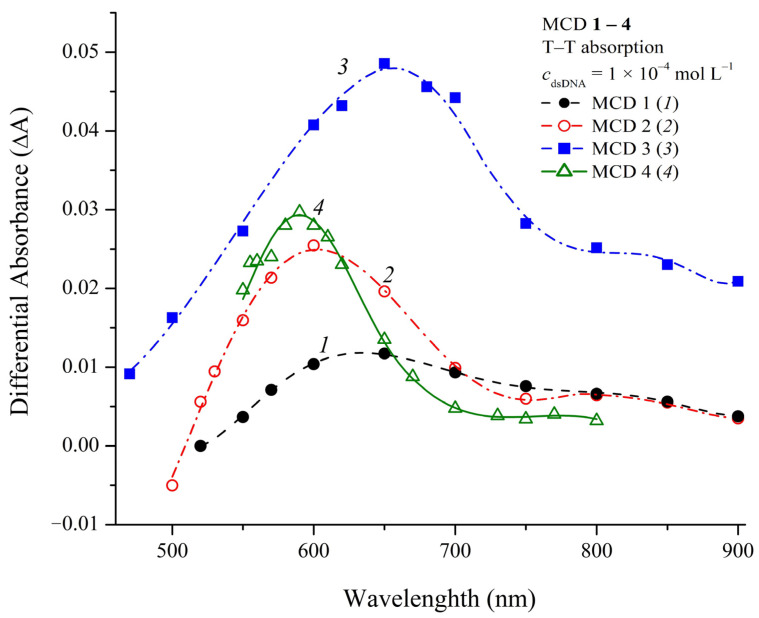
Different T–T absorption spectra of MCD **1**–**4** (*1*–*4*, respectively; *c*_MCD_ = 4–5 × 10^−6^ mol L^−1^) in buffer in the presence of dsDNA (*c*_dsDNA_ = 1 × 10^−4^ mol L^−1^) obtained by flash photolysis at 50 μs after a flash. The T–T absorption spectrum of MCD **4** (at 20 μs after a flash) was obtained by T–T energy transfer from a donor (sodium naphthalene-2-sulfonate, *c*_sens_ = 5 × 10^−6^ mol L^−1^).

**Table 1 ijms-24-13954-t001:** Wavelengths of the maxima of the absorption (λ_abs_), fluorescence (λ_fl_), and fluorescence excitation (λ_ex_) spectra, as well as molar extinction coefficients (ε, L mol^−1^ cm^−1^), of MCD **1**–**4** in some organic solvents and a phosphate buffer (PBS).

MCD	1			2			3			4		
Solvent	λ_abs_	λ_fl_	λ_ex_	λ_abs_	λ_fl_	λ_ex_	λ_abs_	λ_fl_	λ_ex_	λ_abs_	λ_fl_	λ_ex_
	**nm**											
Ethanol	492	526	487	477	496	475	424	455	359, 422	492, 523	568	525
Acetone	492	559	498	478	501	475	424	481	421	493, 523	559	524
DMSO	501	535	496	481	504	479	427	476	424	493, 530	553	536
Acetonitrile	497	555	495	477	501	471	422	457	422	500, 523	548	522
1,4-Dioxane	481	582	478	484	527	472	428	475	425	502, 530	524, 570	484, 517
Chloroform	492	543	491	486	504	481	428	483	428	497, 529	581	526
Ethyl acetate	476	619	459	480	513	471	426	478	426	496, 524	559, 670	484, 516
PBS buffer	491	520	422, 492	478	509	485	422	463–489 ^1^	400, 423	494, 523	512, 554	490, 525
Molar extinction coefficients (ε, L mol^−1^ cm^−1^) ^2^												
Ethanol	58,400			78,800			69,200			49,000		
Acetone	44,300			94,000			50,900			34,000		
DMSO	58,500			60,200			55,100			28,000		
Acetonitrile	48,700			91,200			61,900			44,700		
1,4-Dioxane	19,500			55,400			50,300			28,000		
Chloroform	31,100			85,000			59,200			47,000		
Ethyl acetate	16,000			45,700			44,000			32,600		
PBS buffer	40,000			68,000			61,800			44,500		

^1^ Broad maximum. ^2^ Estimated values determined on the basis of the data of [26] (MCD **1** and **2**) and [42] (MCD **3** and **4**).

**Table 2 ijms-24-13954-t002:** Fluorescence quantum yields (φ_fl_, %), rate constants of radiative (*k*_r_) and nonradiative (*k*_nr_) deactivations, and radiative lifetimes (τ_r_ = 1/*k*_r_) of the excited state of MCD **1**–**4** in ethanol, acetonitrile, and 1,4-dioxane.

Solvent	Ethanol				Acetonitrile				1.4-Dioxane			
Dye	1	2	3	4	1	2	3	4	1	2	3	4
φ_fl_, %	0.11	0.06	0.7	~0.06	0.14	0.02	0.6	~0.05	0.4	0.16	1.3	~0.06
*k*_r_ × 10^−8^, s^−1^	1.9	3.4	4.7	2.3	1.8	3.9	4.1	2.9	0.8	2.4	2.3	1.6
τ_r_, ns	5.2	2.9	2.1	4.4	5.7	2.5	2.4	3.4	12	4.2	4.4	6.2
*k*_nr_ × 10^−11^, s^−1^	1.8	5.7	0.67	3.8	1.2	1.6	0.68	5.0	0.2	1.5	0.17	2.7

**Table 3 ijms-24-13954-t003:** Spectral fluorescent characteristics of MCD **1** and **2** in different organic solvents: wavenumbers of the first moments of the normalized photon distribution (ν_1_, µm^−1^), width of spectral curves (σ, 10^3^ cm^−1^) of the absorption bands, and fluorescence intensity (*I*_fl_, a.u; λ_ex_ = 460 nm) together with some properties of the media: dielectric constant (ε_r_), refractive index (*n*), polarizability (*P**), Reichardt’s *E*_T_(30).

Solvent	ε_r_	*n*	*P**	*E*_T_(30)	MCD 1			MCD 2		
					ν_1_, µm^−1^	σ, 10^3^ cm^−1^	*I*_fl_, a.u	ν_1_, µm^−1^	σ,10^3^ cm^−1^	*I*_fl_, a.u
Acetonitrile	36.2	1.3441	0.2119	45.6	2.095	1.384	0.63	2.158	1.092	0.24
Acetone	20.7	1.3586	0.2199	42.2	2.076	1.194	1.12	2.146	1.015	0.23
Ethanol	24.3	1.3614	0.2215	51.9	2.086	1.188	0.49	2.159	1.102	0.56
Ethyl acetate	6.02	1.3724	0.2275	38.1	2.104	1.526	0.46	2.144	1.043	0.42
Isopentyl alcohol	15.2	1.4075	0.2275	49	2.126	1.472	2.17	2.138	0.996	1.32
1.4-Dioxane	2.22	1.4224	0.2464	36	2.114	1.322	1.09	2.127	1.064	1.39
Chloroform	4.81	1.4459	0.2586	39.1	2.069	0.977	0.58	2.122	1.006	0.69
DMSO	49	1.4793	0.2666	45.1	2.039	1.073	0.36	2.136	1.095	1.06
Benzyl alcohol	13.5	1.5396	0.2837	50.4	2.026	1.066	0.98	2.117	1.046	2.39

**Table 4 ijms-24-13954-t004:** Spectral fluorescent characteristics of MCD **1**–**4** in the presence of dsDNA, ssDNA and HSA: absorption maxima (λ_abs_^b^, nm); wavenumbers of the first moments of the normalized photon distribution (ν_1_, µm^−1^) and corresponding position of the band (*M*^−1^, nm); absorption bandwidths (σ, cm^−1^) and their relative changes (Δσ, %); as well as the fluorescence (λ_fl_^b^, nm) and fluorescence excitation (λ_ex_^b^, nm) maxima; the corresponding shifts of the spectral maxima (Δλ_abs,_ Δ*M*^−1^, Δλ_fl_, nm); and fluorescence quantum yields of the dyes (φ_fl_^b^, %).

MCD	λ_abs_^b^	Δλ_abs_	ν_1_	*M* ^−1^	Δ*M*^−1^	σ	Δσ	λ_fl_^b^	Δλ_fl_	λ_ex_^b^	φ_fl_^b^
	nm		cm^−1^	nm		cm^−1^	%	nm			%
*c*_dsDNA_~5 × 10^−5^ mol L^−1^											
**1**	503	12	20,450	489	8	1100	−4	525	5	505	13 ± 1.5
**2**	485	7	21,146	473	10	1130	−3	505	−4	485	14 ± 1.7
**3**	423	1	24,030	416	6	1170	−2	460	−16	430	4.8 ± 0.5
**4**	522	−1	19,880	503	0	1242	1	522	10	506	~1.5 ± 0.2
*c*_ssDNA_~5 × 10^−5^ mol L^−1^											
**1**	499	8	20,550	486	5	1074	−7	525	5	507	12 ± 1.5
**2**	480	2	21,490	465	2	1088	−6	508	2	485	14 ± 1.7
**3**	424	2	23,940	418	8	1060	−11	460	−16	430	4.7 ± 0.5
**4**	526	3	19,644	509	6	1038	−16	508	−4	488	~1.4 ± 0.2
*c*_HSA_~5 × 10^−5^ mol L^−1^											
**1**	493	2	20,756	482	1	1101	−4	523	3	498	~1 ± 0.1
**2**	478	0	21,600	463	0	1278	10	504	−5	486	4 ± 0.8
**3**	379, 423	2	24,760	404	−6	1450	22	472	−4	360, 427	~50 ± 6
**4**	498, 521	2	20,050	499	−4	1988	61	515	3	362	~0.5 ± 0.1

**Table 5 ijms-24-13954-t005:** Results of molecular docking of MCD **1**−**4** with dsDNA and HSA: the number of runs; free energy of binding (Δ*G*_est_, kcal mol^−1^, at 25 °C); intermolecular interaction energy (*E*_intmol_, kcal mol^−1^); sum of van der Waals interaction, hydrogen bonding, and desolvation energies (*E*_VdWHD_, kcal mol^−1^); electrostatic interaction energy (*E*_el_, kcal mol^−1^); total internal energy (*E*_tint_, kcal mol^−1^); torsional free energy (*E*_tor_, kcal mol^−1^).

MCD	Run	ΔG_est_	*E* _intmol_	*E* _VdWHD_	*E* _el_	*E* _tint_	*E* _tor_
		kcal mol^−1^					
DNA (minor groove)							
**1**	1	−10.18	−12.57	−11.28	−1.29	−0.65	+2.39
**2**	3	−9.01	−11.40	−11.39	−0.01	−1.28	+2.39
**3**	10	−9.70	−10.59	−10.56	−0.04	−0.04	+0.89
**4**	4	−10.40	−11.30	−11.22	−0.08	−0.62	+0.89
DNA (intercalation)							
**1**	4	−8.07	−10.46	−10.16	−0.3	−1.14	+2.39
**2**	6	−8.58	−10.96	−10.86	−0.11	−1.88	+2.39
**3**	6	−7.83	−8.73	−8.63	−0.09	−0.75	+0.89
**4**	3	−8.40	−8.70	−8.67	−0.03	−0.13	+0.30
HSA (Sudlow I)							
**1**	2	−7.85	−10.24	−10.15	−0.09	−0.88	+2.39
**2**	2	−8.79	−11.18	−11.08	−0.1	−1.19	+2.39
**3**	5	−7.47	−8.36	−8.33	-0.04	−0.73	+0.89
**4**	1	−8.20	−9.09	−9.09	−0.01	−0.90	+0.89
HSA (Sudlow II)							
**1**	5	−7.07	−9.45	−9.17	−0.28	−0.90	+2.39
**2**	2	−7.44	−9.82	−9.65	−0.18	−1.45	+2.39
**3**	8	−5.75	−6.65	−6.57	−0.08	−0.75	+0.89
**4**	8	−7.61	−8.51	−8.52	+0.01	−0.91	+0.89

**Table 6 ijms-24-13954-t006:** Parameters of the Lehrer–Leavis model for quenching of HSA fluorescence by MCD, measured using SFS: *K*_SV_ and α (Equation (4)).

Fluorophore/ SFS Δλ, nm	MCD			
	1	2	3	4
	*K*_SV_, L mol^−1^; α			
Tyr/15 nm	2.18 × 10^4^; 0.61	2.16 × 10^4^; 0.45	1.12 × 10^5^; 0.15	3.82 × 10^4^; 0.33
Trp/60 nm	4.04 × 10^4^; 0.40	2.53 × 10^4^; 0.55	6.12 × 10^7^; 0.82	2.10 × 10^4^; 0.60

**Table 7 ijms-24-13954-t007:** Relative changes in molar extinction coefficients on complexation (ε_f_/ε_b_), effective constants of the complexation reaction (*K*_effBH_, *K*_effHill_, *K*_1Sc_, *K*_2Sc_; L mol^−1^) of MCD **1**–**4** with biomolecules (dsDNA, ssDNA, HSA) in PBS buffer determined by fluorescent titration. Presented are also the proportions of bound dye (α_b_, at *c*_biomol_ = 5 ×10^−5^ mol L^−1^), the fluorescence growth at 100% binding *F*_max_/*F*_0_ (from the Benesi–Hildebrand dependence), Hill coefficients (*n*_Hill_) and stoichiometric coefficients (*n*_Sc_) from the Scatchard equations, linear portions of the fluorescence intensity vs. concentration plots (LR, µmol L^−1^), and detection limits LOD/LOQ (mol L^−1^).

MCD	ε_f_/ε_b_	*K*_effBH_ ×10^−4^, L mol^−1^	$\frac{F_{max}}{F_{0}}$	α_b_, ^1^ %	*K*_effHill_ ×10^−4^, L mol^−1^	*n* _Hill_	*K* _1Sc_	*K* _2Sc_	*n* _Sc_	LR, µmol L^−1^	LOD	LOQ
							×10^−5^, L mol^−1^				× 10^8^, mol L^−1^	
dsDNA												
**1**	1.05	5.4	460	80%	5.34	1.04	7.93	1.69	2	0–3	0.72	2.1
**2**	1.36	3.94	198	72%	3.8	0.99	4.35	24.5	56	0–~2	1.2	3.5
**3**	3.90	1.35	136	40%	0.69	0.88	1.35	4.8	35	0–25	4.1	12.6
**4**	1.83	1.95	169	46%	0.185	0.62	1.86	0.37	2	1.5–13 ^2^	9.4	21
ssDNA												
**1**	1.73	2.42	60.9	60%	0.283	0.7	2.28	1.59	2	0–3.5	4.0	12
**2**	0.72	1.64	178	54%	0.31	0.95	1.86	3.9	21	0–~12	10.7	32
**3**	2.88	1.16	71	37%	0.294	0.76	1.12	4.3	37	0–10	8.7	26
**4**	–	0.68	61	25%	0.035	0.57	0.65	0.195	3	1–140 ^2^	76	230
HSA												
**1**	1.59	0.80	65	27%	0.335	0.89	–	0.073	1	0–0.6 ^3^	27	89
**2**	2.19	6.34	47	74%	6.33	0.95	–	0.60		0–10	3.4	11.3
**3**	–	1.76	26	50%	2.78	0.59	–	0.34		0–1	7.7	23
**4**	–	0.482	109	20%	0.183	0.61	–	0.17		0.7–7.5 ^2^	73	220

^1^ At *c*_biomol_ = 5 × 10^−5^ mol L^−1^. ^2^ The dependence is nonlinear (the initial portion is excluded). ^3^ The dependence is nonlinear (the initial linear portion is presented).

## Data Availability

The data that support the findings of this study are available from the authors, P.G.P. and A.S.T., upon reasonable request.

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
