# Peer review of "Photonics of Some Monomethine Cyanine Dyes in Solutions and in Complexes with Biomolecules"

_ijms, 2023, doi:10.3390/ijms241813954_

Round 1
Reviewer 1 Report
Comments and Suggestions for Authors
The manuscript ijms-2577122, titled “Photonics of some monomethine cyanine dyes in solutions and in complexes with biomolecules”, describes the spectroscopic investigation and docking of 4 symetric and non-symmetric monomethine cyanine dyes in various organic solvents, in aqueous buffer solutions, and in the presence of DNA and HSA. Overall, this is an original study on an important research topic.
Nevertheless, in my opinion, there are several issues that should be addressed before being considered for publication in the International Journal of Molecular Sciences.
Remarks and questions:
1. Introduction: "MCD are also used in real-time PCR analysis of nucleic acids." - Please include a reference here
1. Introduction: "In search of new fluorescent probes for nucleic acids, a number of unsymmetrical MCD were synthesized, which interacted with RNA and DNA with fluorescence growth" - please include more references here.
1. Introduction: "In the present work, we have undertaken a comprehensive study of the spectral-fluorescent properties of two of the previously synthesized unsymmetrical MCD, 1 and 2, in various solvents, their noncovalent interaction with dsDNA, ssDNA, and human serum albumin (HSA), and photonics (spectral-fluorescent and photochemical properties) of dye–biomolecule complexes with a view to the possible use of these dyes as probes in biomolecular systems. Symmetrical MCD 3 and 4, having terminal hetero cycles similar to those of 1 and 2, were also studied for comparison" - Please include references for all known dyes (some of them are shown in the list bellow).
1. Introduction, Page 2: "In search of new fluorescent probes for nucleic acids, a number of unsymmetrical MCD were synthesized, which interacted with RNA and DNA with fluorescence growth [13]. However, their photophysical properties were studied very briefly" - Over the past 30 years, a plethora of literature sources can be found on the use of monomethine cyanine dyes as nucleic acid probes, as well as a detailed investigation of their photophysical properties. The authors are advised to include more related references aiming to give a better overview on this topic. Out of the 65 references, 12 are on publications of current authors. Thus, the list could further be extended with more recent papers, including works from other research groups working in the same field with structurally similar compounds. Based on a Reaxys and Scifinder research, some suggestions on recent and original literature sources to be added are listed below:
Molecules. 27 (2022) 5779. https://doi.org/10.3390/molecules27185779
Journal of Fluorescence 12, 225–229 (2002). https://doi.org/10.1023/A:1016817018665
New J. Chem. 45 (2021) 12818–12829. https://doi.org/10.1039/D1NJ01659H
CHIMIA. 71 (2017) 26–26. https://doi.org/10.2533/chimia.2017.26
Journal of Photochemistry and Photobiology A: Chemistry. 397 (2020) 112598. https://doi.org/10.1016/j.jphotochem.2020.112598
Journal of Molecular Liquids. 302 (2020) 112569. https://doi.org/10.1016/j.molliq.2020.112569
Dyes Pigm. 148 (2018) 452–459. https://doi.org/10.1016/j.dyepig.2017.09.049
Bioorganic & Medicinal Chemistry. 17 (2009) 585–591. https://doi.org/10.1016/j.bmc.2008.11.083
Dyes Pigm. 157 (2018) 267–277. https://doi.org/10.1016/j.dyepig.2018.04.064
Cyanine dyes as fluorescent non-covalent labels for nucleic acid research, in: Functional Dyes, Elsevier, 2006: pp. 137–183
Methods Appl. Fluoresc. 9 (2021) 045002. https://doi.org/10.1088/2050-6120/ac10ad
J. Am. Chem. Soc. 116 (1994) 8459–8465. https://doi.org/10.1021/ja00098a004
J. Phys. Chem. 98 (1994) 10313–10321. https://doi.org/10.1021/j100091a055
Biomolecules. 13 (2023) 128. https://doi.org/10.3390/biom13010128
Journal of Fluorescence. 12 (2002) 225–229. https://doi.org/10.1023/A:1016817018665
Nucleic Acids Research. 44 (2016) 3971–3988. https://doi.org/10.1093/nar/gkw237
J. Phys. Chem. B. 121 (2017) 10242–10248. https://doi.org/10.1021/acs.jpcb.7b08187
J. Phys. Chem. B. 108 (2004) 4268–4274. https://doi.org/10.1021/jp035617s
Heterocyclic Communications. 19 (2013) 1–11. https://doi.org/10.1515/hc-2013-0012
Environ. Sci. Technol. 1 (1967) 57–65. https://doi.org/10.1021/es60001a006.
Phys. Chem. Chem. Phys. 2 (2000) 4784–4792. https://doi.org/10.1039/B004637J
Zeitschrift Für Naturforschung B. 48 (1993) 461–470. https://doi.org/10.1515/znb-1993-0411
Bulletin des Sociétés Chimiques Belges. 73 (1964) 921–943. https://doi.org/10.1002/bscb.19640731110
J. Chem. Soc. (1930) 2502–2510. https://doi.org/10.1039/JR9300002502
J. Chem. Soc. (1949) 1503–1509. https://doi.org/10.1039/JR9490001503
NATIONAL UNIVERSITY OF SINGAPORE - WO2019/177541, 2019, A1
AGENCY FOR SCIENCE, TECHNOLOGY AND RESEARCH - US2016/195519, 2016, A1
G HOLDINGS INC; I.G. FARBENINDUSTRIE AG (historic) - US2292021, 1933, A
EASTMAN KODAK CO - US2241237, 1936, A
The term in silico should be given in italics.
Scheme 1 should be renamed to Figure 1, while other figures need to be renumbered accordingly.
Scheme 1: The counterions must be shown for each chemical structure since the type of anion could have a substantial impact on the photophysical properties of a given luminophore.
Figure 1: The y-axis on the right-hand side (fluorescence intensity) needs to be renamed to either normalized or the scale needs to be added.
Figure 1 - Please include dye concentration in the figure caption. Also, the Y axis (Absorbance) should be expressed either as a molar extinction coefficient or normalized. The solvents could also be shown in the figure legend.
Figures 1-4: Although there is a description of the conditions in the figure caption, it is somehow confusing for the readers and not easy to follow. Thus, I would suggest the authors to include this information on the figures themselves as legends.
Table 1 - The authors are advised to include molar extinction coefficients of the dyes under investigation.
Reichardt's ET(30) - Please include references for the presented values.
Tables 2, 3, 5, and 7 - Please mind significant figures and apply corrections where necessary.
3. Materials and methods: Please include the size of cells used for measuring the absorption and emission spectra.
3. Materials and methods: The fluorescence quantum yield was evaluated relative to fluorescein (Φfl.st. = 0.93% in 0.1 M NaOH (aq) [56]). This was presumably calculated based on the formula S1 provided in the SI material file. On the other hand, according to the formula description, nst. and n are the refractive indices of the standard and dye solutions. What is the refractive index of 0.1 M NaOH (aq) solution? Please incude a reference where applicable.
The use of English was found appropriate.
Reviewer 2 Report
Comments and Suggestions for Authors
1. In scheme 1 the counter ions are not included. Please, provide the counter ions of the dyes cations.
2. Figure 1 is messed up and doesn't look fit for publication. Moreover, the absorption in this figure has maximum values of 0.15 au, which is far from the accuracy of the method. Please modify the figure, repeat the measurements at absorption values around 1 au and make figure 1 clearer for the readers. Otherwise clearly explain the reasons to work in so low concentrations.
3. In Figure 2A the absorbance is below 0.1, please repeat the experiments with higher concentrations or explain why you have to work with such dilute solutions. Considering the high molar absorptivity of this type of dyes, the absorbance you get is typical for concentrations around 10-6 M. Please write the exact concentrations of the dyes in your working solutions below the figures.
4. In Figure 2B and 2C, fluorescence has no values. Please give values in the ordinate of the figures.
5. The same thing I mentioned about Figure 2 applies to Figures 3 and 4.
The work is of interest to the readers and deserves to be published in the present journal.
Comments on the Quality of English LanguageThe English needs to be improved.
Round 2
Reviewer 1 Report
Comments and Suggestions for Authors
The authors have completed the revision part of their submitted work and followed all relative suggestions aiming to improve the manuscript. Hence, I believe it should be published in its current form.
Reviewer 2 Report
Comments and Suggestions for Authors
The work is of interest to the readers and deserves to be published in the present journal.